# Expectation Propagation for t-Exponential Family Using q-Algebra

**Futoshi Futami**
The University of Tokyo, RIKEN
futami@ms.k.u-tokyo.ac.jp

**Issei Sato**
The University of Tokyo, RIKEN
sato@k.u-tokyo.ac.jp

**Masashi Sugiyama**
RIKEN, The University of Tokyo
sugi@k.u-tokyo.ac.jp

## Abstract

Exponential family distributions are highly useful in machine learning since their calculation can be performed efficiently through natural parameters. The exponential family has recently been extended to the *t-exponential family*, which contains Student-t distributions as family members and thus allows us to handle noisy data well. However, since the t-exponential family is defined by the *deformed exponential*, an efficient learning algorithm for the t-exponential family such as expectation propagation (EP) cannot be derived in the same way as the ordinary exponential family. In this paper, we borrow the mathematical tools of *q-algebra* from statistical physics and show that the *pseudo additivity* of distributions allows us to perform calculation of t-exponential family distributions through natural parameters. We then develop an *expectation propagation* (EP) algorithm for the t-exponential family, which provides a deterministic approximation to the posterior or predictive distribution with simple moment matching. We finally apply the proposed EP algorithm to the *Bayes point machine* and *Student-t process classification*, and demonstrate their performance numerically.

## 1 Introduction

Exponential family distributions play an important role in machine learning, due to the fact that their calculation can be performed efficiently and analytically through natural parameters or expected sufficient statistics [1]. This property is particularly useful in the Bayesian framework since a conjugate prior always exists for an exponential family likelihood and the prior and posterior are often in the same exponential family. Moreover, parameters of the posterior distribution can be evaluated only through natural parameters.

As exponential family members, Gaussian distributions are most commonly used because their moments, conditional distribution, and joint distribution can be computed analytically. Gaussian processes are a typical Bayesian method based on Gaussian distributions, which are used for various purposes such as regression, classification, and optimization [8]. However, Gaussian distributions are sensitive to outliers and heavier-tailed distributions are often more preferred in practice. For example, Student-t distributions and Student-t processes are good alternatives to Gaussian distributions [4] and Gaussian processes [10], respectively.

A technical problem of the Student-t distribution is that it does not belong to the exponential family unlike the Gaussian distribution and thus cannot enjoy good properties of the exponential family. To cope with this problem, the exponential family was recently generalized to the *t-exponential family* [3],

which contains Student-t distributions as family members. Following this line, the Kullback-Leibler divergence was generalized to the *t-divergence*, and approximation methods based on t-divergence minimization have been explored [2]. However, the t-exponential family does not allow us to employ standard useful mathematical tricks, e.g., logarithmic transformation does not reduce the product of t-exponential family functions into summation. For this reason, the t-exponential family unfortunately does not inherit an important property of the original exponential family, that is, calculation can be performed through natural parameters. Furthermore, while the dimensionality of sufficient statistics is the same as that of the natural parameters in the exponential family and thus there is no need to increase the parameter size to incorporate new information [9], this useful property does not hold in the t-exponential family.

The purpose of this paper is to further explore mathematical properties of natural parameters of the t-exponential family through *pseudo additivity* of distributions based on *q-algebra* used in statistical physics [7, 11]. More specifically, our contributions in this paper are three-fold:

1. We show that, in the same way as ordinary exponential family distributions, q-algebra allows us to handle the calculation of t-exponential family distributions through natural parameters.

2. Our q-algebra based method enables us to extend *assumed density filtering* (ADF) [2] and develop an algorithm of *expectation propagation* (EP) [6] for the t-exponential family. In the same way as the original EP algorithm for ordinary exponential family distributions, our EP algorithm provides a deterministic approximation to the posterior or predictive distribution for t-exponential family distributions with simple moment matching.

3. We apply the proposed EP algorithm to the *Bayes point machine* [6] and *Student-t process classification*, and we demonstrate their usefulness as alternatives to the Gaussian approaches numerically.

## 2 t-exponential Family

In this section, we review the *t-exponential family* [3, 2], which is a generalization of the exponential family.

The t-exponential family is defined as,

$$p(x; \theta) = \exp_t(\langle \Phi(x), \theta \rangle - g_t(\theta)), \tag{1}$$

where $\exp_t(x)$ is the *deformed exponential function* defined as

$$\exp_t(x) = \begin{cases} \exp(x) & \text{if } t = 1, \\ [1 + (1-t)x]^{\frac{1}{1-t}} & \text{otherwise,} \end{cases} \tag{2}$$

and $g_t(\theta)$ is the log-partition function that satisfies

$$\nabla_\theta g_t(\theta) = \mathbb{E}_{p^{\text{es}}}[\Phi(x)]. \tag{3}$$

The notation $\mathbb{E}_{p^{\text{es}}}$ denotes the expectation over $p^{\text{es}}(x)$, where $p^{\text{es}}(x)$ is the *escort distribution* of $p(x)$ defined as

$$p^{\text{es}}(x) = \frac{p(x)^t}{\int p(x)^t \mathrm{d}x}. \tag{4}$$

We call $\theta$ a *natural parameter* and $\Phi(x)$ *sufficient statistics*.

Let us express the $k$-dimensional Student-t distribution with $v$ degrees of freedom as

$$\text{St}(x; v, \mu, \Sigma) = \frac{\Gamma((v+k)/2)}{(\pi v)^{k/2} \Gamma(v/2) |\Sigma|^{1/2}} \left( 1 + (x-\mu)^\top (v\Sigma)^{-1} (x-\mu) \right)^{-\frac{v+k}{2}}, \tag{5}$$

where $\Gamma(x)$ is the gamma function, $|A|$ is the determinant of matrix $A$, and $A^\top$ is the transpose of matrix $A$. We can confirm that the Student-t distribution is a member of the t-exponential family as follows. First, we have

$$\text{St}(x; v, \mu, \Sigma) = \left( \Psi + \Psi \cdot (x-\mu)^\top (v\Sigma)^{-1} (x-\mu) \right)^{\frac{1}{1-t}}, \tag{6}$$

$$\text{where } \Psi = \left( \frac{\Gamma((v+k)/2)}{(\pi v)^{k/2} \Gamma(v/2) |\Sigma|^{1/2}} \right)^{1-t}. \tag{7}$$

Note that relation $-(v+k)/2 = 1/(1-t)$ holds, from which we have

$$\langle \Phi(x), \theta \rangle = \left( \frac{\Psi}{1-t} \right) (x^\top K x - 2\mu^\top K x), \tag{8}$$

$$g_t(\theta) = - \left( \frac{\Psi}{1-t} \right) (\mu^\top K \mu + 1) + \frac{1}{1-t}, \tag{9}$$

where $K = (v\Sigma)^{-1}$. Then, we can express the Student-t distribution as a member of the t-exponential family as:

$$\mathrm{St}(x; v, \mu, \Sigma) = \left( 1 + (1-t)\langle \Phi(x), \theta \rangle - g_t(\theta) \right)^{\frac{1}{1-t}} = \exp_t \left( \langle \Phi(x), \theta \rangle - g_t(\theta) \right). \tag{10}$$

If $t = 1$, the deformed exponential function is reduced to the ordinary exponential function, and therefore the t-exponential family is reduced to the ordinary exponential family, which corresponds to the Student-t distribution with infinite degrees of freedom. For t-exponential family distributions, the *t-divergence* is defined as follows [2]:

$$D_t(p\|\widetilde{p}) = \int \left( p^{\mathrm{es}}(x) \ln_t p(x) - p^{\mathrm{es}}(x) \ln_t \widetilde{p}(x) \right) \mathrm{d}x, \tag{11}$$

where $\ln_t x := \frac{x^{1-t}-1}{1-t}$ $(x \geq 0, t \in \mathbb{R}^+)$ and $p^{\mathrm{es}}(x)$ is the escort function of $p(x)$.

# 3    Assumed Density Filtering and Expectation Propagation

We briefly review the assumed density filtering (ADF) and expectation propagation (EP) [6].

Let $D = \{(x_1, y_1), \ldots, (x_i, y_i)\}$ be input-output paired data. We denote the likelihood for the $i$-th data as $l_i(w)$ and the prior distribution of parameter $w$ as $p_0(w)$. The total likelihood is given as $\prod_i l_i(w)$ and the posterior distribution can be expressed as $p(w|D) \propto p_0(w) \prod_i l_i(w)$.

## 3.1    Assumed Density Filtering

ADF is an online approximation method for the posterior distribution.

Suppose that $i - 1$ samples $(x_1, y_1), \ldots, (x_{i-1}, y_{i-1})$ have already been processed and an approximation to the posterior distribution, $\widetilde{p}_{i-1}(w)$, has already been obtained. Given the $i$-th sample $(x_i, y_i)$, the posterior distribution $p_i(w)$ can be obtained as

$$p_i(w) \propto \widetilde{p}_{i-1}(w) l_i(w). \tag{12}$$

Since the true posterior distribution $p_i(w)$ cannot be obtained analytically, it is approximated in ADF by minimizing the Kullback-Leibler (KL) divergence from $p_i(w)$ to its approximation:

$$\widetilde{p}_i = \arg \min_{\widetilde{p}} \mathrm{KL}(p_i\|\widetilde{p}). \tag{13}$$

Note that if $p_i$ and $\widetilde{p}$ are both exponential family members, the above calculation is reduced to moment matching.

## 3.2    Expectation Propagation

Although ADF is an effective method for online learning, it is not favorable for non-online situations, because the approximation quality depends heavily on the permutation of data [6]. To overcome this problem, EP was proposed [6].

In EP, an approximation of the posterior that contains whole data terms is prepared beforehand, typically as a product of data-corresponding terms:

$$\widetilde{p}(w) = \frac{1}{Z} \prod_i \widetilde{l}_i(w), \tag{14}$$

where $Z$ is the normalizing constant. In the above expression, factor $\widetilde{l}_i(w)$, which is often called a *site approximation* [9], corresponds to the local likelihood $l_i(w)$. If each $\widetilde{l}_i(w)$ is an exponential family member, the total approximation also belongs to the exponential family.

Differently from ADF, EP has these site approximation with the following four steps, which is iteratively updated. First, when we update site $\widetilde{l}_j(w)$, we eliminate the effect of site $j$ from the total approximation as

$$\widetilde{p}^{\backslash j}(w) = \frac{\widetilde{p}(w)}{\widetilde{l}_j(w)}, \tag{15}$$

where $\widetilde{p}^{\backslash j}(w)$ is often called a *cavity distribution* [9]. If an exponential family distribution is used, the above calculation is reduced to subtraction of natural parameters. Second, we incorporate likelihood $l_j(w)$ by minimizing the divergence $\mathrm{KL}(\widetilde{p}^{\backslash j}(w)l_j(w)/Z^{\backslash j} \| \widetilde{p}(w))$, where $Z^{\backslash j}$ is the normalizing constant. Note that this minimization is reduced to moment matching for the exponential family. After this step, we obtain $\widetilde{p}(w)$. Third, we exclude the effect of terms other than $j$, which is equivalent to calculating a cavity distribution as $\widetilde{l}_j(w)^{\mathrm{new}} \propto \frac{\widetilde{p}(w)}{\widetilde{p}^{\backslash j}(w)}$. Finally, we update the site approximation by replacing $\widetilde{l}_j(w)$ by $\widetilde{l}_j(w)^{\mathrm{new}}$.

Note again that calculation of EP is reduced to addition or subtraction of natural parameters for the exponential family.

### 3.3 ADF for t-exponential Family

ADF for the t-exponential family was proposed in [2], which uses the *t-divergence* instead of the KL divergence:

$$\widetilde{p} = \arg\min_{p'} D_t(p\|p') = \int \left( p^{\mathrm{es}}(x) \ln_t p(x) - p^{\mathrm{es}}(x) \ln_t p'(x;\theta) \right) \mathrm{d}x. \tag{16}$$

When an approximate distribution is chosen from the t-exponential family, we can utilize the property $\nabla_\theta g_t(\theta) = \mathbb{E}_{\widetilde{p^{\mathrm{es}}}}(\Phi(x))$, where $\widetilde{p^{\mathrm{es}}}$ is the escort function of $\widetilde{p}(x)$. Then, minimization of the t-divergence yields

$$\mathbb{E}_{p^{\mathrm{es}}}[\Phi(x)] = \mathbb{E}_{\widetilde{p^{\mathrm{es}}}}[\Phi(x)]. \tag{17}$$

This is moment matching, which is a celebrated property of the exponential family. Since the expectation is taken with respect to the escort function, this is called *escort moment matching*.

As an example, let us consider the situation where the prior is the Student-t distribution and the posterior is approximated by the Student-t distribution: $p(w|D) \cong \widetilde{p}(w) = \mathrm{St}(w; \widetilde{\mu}, \widetilde{\Sigma}, v)$. Then the approximated posterior $\widetilde{p}_i(w) = \mathrm{St}(w; \widetilde{\mu}^{(i)}, \widetilde{\Sigma}^i, v)$ can be obtained by minimizing the t-divergence from $p_i(w) \propto \widetilde{p}_{i-1}(w)\widetilde{l}_i(w)$ as

$$\arg\min_{\mu', \Sigma'} D_t(p_i(w) \| \mathrm{St}(w; \mu', \Sigma', v)). \tag{18}$$

This allows us to obtain an analytical update expression for t-exponential family distributions.

## 4 Expectation Propagation for t-exponential Family

As shown in the previous section, ADF has been extended to EP (which resulted in moment matching for the exponential family) and to the t-exponential family (which yielded escort moment matching for the t-exponential family). In this section, we combine these two extensions and propose EP for the t-exponential family.

### 4.1 Pseudo Additivity and q-Algebra

Differently from ordinary exponential functions, *deformed* exponential functions do not satisfy the product rule:

$$\exp_t(x) \exp_t(y) \neq \exp_t(x + y). \tag{19}$$

For this reason, the cavity distribution cannot be computed analytically for the t-exponential family.

On the other hand, the following equality holds for the deformed exponential functions:

$$\exp_t(x)\exp_t(y) = \exp_t(x + y + (1 - t)xy), \tag{20}$$

which is called *pseudo additivity*.

In statistical physics [7, 11], a special algebra called *q-algebra* has been developed to handle a system with pseudo additivity. We will use the q-algebra for efficiently handling t-exponential distributions.

**Definition 1 (q-product)** *Operation* $\otimes_q$ *called the* q-product *is defined as*

$$x \otimes_q y := \begin{cases} [x^{1-q} + y^{1-q} - 1]^{\frac{1}{1-q}} & \text{if } x > 0, y > 0, x^{1-q} + y^{1-q} - 1 > 0, \\ 0 & \text{otherwise.} \end{cases} \tag{21}$$

**Definition 2 (q-division)** *Operation* $\oslash_q$ *called the* q-division *is defined as*

$$x \oslash_q y := \begin{cases} [x^{1-q} - y^{1-q} - 1]^{\frac{1}{1-q}} & \text{if } x > 0, y > 0, x^{1-q} - y^{1-q} - 1 > 0, \\ 0 & \text{otherwise.} \end{cases} \tag{22}$$

**Definition 3 (q-logarithm)** *The* q-logarithm *is defined as*

$$\ln_q x := \frac{x^{1-q} - 1}{1 - q} \quad (x \geq 0, q \in \mathbb{R}^+). \tag{23}$$

The q-division is the inverse of the q-product (and visa versa), and the q-logarithm is the inverse of the q-exponential (and visa versa). From the above definitions, the q-logarithm and q-exponential satisfy the following relations:

$$\ln_q(x \otimes_q y) = \ln_q x + \ln_q y, \tag{24}$$

$$\exp_q(x) \otimes_q \exp_q(y) = \exp_q(x + y), \tag{25}$$

which are called the *q-product rules*. Also for the q-division, similar properties hold:

$$\ln_q(x \oslash_q y) = \ln_q x - \ln_q y, \tag{26}$$

$$\exp_q(x) \oslash_q \exp_q(y) = \exp_q(x - y), \tag{27}$$

which are called the *q-division rules*.

## 4.2 EP for t-exponential Family

The q-algebra allows us to recover many useful properties from the ordinary exponential family. For example, the q-product of t-exponential family distributions yields an unnormalized t-exponential distribution:

$$\exp_t(\langle \Phi(x), \theta_1 \rangle - g_t(\theta_1)) \otimes_t \exp_t(\langle \Phi(x), \theta_2 \rangle - g_t(\theta_2))$$
$$= \exp_t(\langle \Phi(x), (\theta_1 + \theta_2) \rangle - \widetilde{g}_t(\theta_1, \theta_2)). \tag{28}$$

Based on this q-product rule, we develop EP for the t-exponential family.

Consider the situation where prior distribution $p^{(0)}(w)$ is a member of the t-exponential family. As an approximation to the posterior, we choose a t-exponential family distribution

$$\widetilde{p}(w; \theta) = \exp_t(\langle \Phi(w), \theta \rangle - g_t(\theta)). \tag{29}$$

In the original EP for the ordinary exponential family, we considered an approximate posterior of the form

$$\widetilde{p}(w) \propto p^{(0)}(w) \prod_i \widetilde{l}_i(w), \tag{30}$$

that is, we factorized the posterior to a product of site approximations corresponding to data. On the other hand, in the case of the t-exponential family, we propose to use the following form called the *t-factorization*:

$$\widetilde{p}(w) \propto p^{(0)}(w) \otimes_t \prod_i \otimes_t \widetilde{l}_i(w). \tag{31}$$

The t-factorization is reduced to the original factorization form when $t = 1$.

This t-factorization enables us to calculate EP update rules through natural parameters for the t-exponential family in the same way as the ordinary exponential family. More specifically, consider the case where factor $j$ of the t-factorization is updated in four steps in the same way as original EP.

(I) First, we calculate the cavity distribution by using the q-division as

$$\widetilde{p}^{\backslash j}(w) \propto \widetilde{p}(w) \oslash_t \widetilde{l}_j(w) \propto p^{(0)}(w) \otimes_t \prod_{i \neq j} \otimes_t \widetilde{l}_i(w). \tag{32}$$

The above calculation is reduced to subtraction of natural parameters by using the q-algebra rules:

$$\theta^{\backslash j} = \theta - \theta^{(j)}. \tag{33}$$

(II) The second step is inclusion of site likelihood $l_j(w)$, which can be performed by $\widetilde{p}^{\backslash j}(w) l_j(w)$. The site likelihood $l_j(w)$ is incorporated to approximate the posterior by the ordinary product not the q-product. Thus moment matching is performed to obtain a new approximation. For this purpose, the following theorem is useful.

**Theorem 1** *The expected sufficient statistic,*

$$\eta = \nabla_\theta g_t(\theta) = \mathbb{E}_{\widetilde{p^{es}}}[\Phi(w)], \tag{34}$$

*can be derived as*

$$\eta = \eta^{\backslash j} + \frac{1}{Z_2} \nabla_{\theta^{\backslash j}} Z_1, \tag{35}$$

$$\text{where} \quad Z_1 = \int \widetilde{p}^{\backslash j}(w)(l_j(w))^t \mathrm{d}w, \quad Z_2 = \int \widetilde{p^{es}}^{\backslash j}(w)(l_j(w))^t \mathrm{d}w. \tag{36}$$

A proof of Theorem 1 is given in Appendix A of the supplementary material. After moment matching, we obtain an approximation, $\widetilde{p}_{\text{new}}(w)$.

(III) Third, we exclude the effect of sites other than $j$. This is achieved by

$$\widetilde{l}_j^{\text{new}}(w) \propto \widetilde{p}_{\text{new}}(w) \oslash_t \widetilde{p}^{\backslash j}(w), \tag{37}$$

which is reduced to subtraction of natural parameter

$$\theta_{\text{new}}^{\backslash j} = \theta^{\text{new}} - \theta^{\backslash j}. \tag{38}$$

(IV) Finally, we update the site approximation by replacing $\widetilde{l}_j(w)$ with $\widetilde{l}_j(w)^{\text{new}}$.

These four steps are our proposed EP method for the t-exponential family. As we have seen, these steps are reduced to the ordinary EP steps if $t = 1$. Thus, the proposed method can be regarded as an extention of the original EP to the t-exponential family.

### 4.3 Marginal Likelihood for t-exponential Family

In the above, we omitted the normalization term of the site approximation to simplify the derivation. Here, we derive the marginal likelihood, which requires us to explicitly take into account the normalization term $\widetilde{C}_i$:

$$\widetilde{l}_i(w|\widetilde{C}_i, \widetilde{\mu}_i, \widetilde{\sigma}_i^2) = \widetilde{C}_i \otimes_t \exp_t(\langle \Phi(w), \theta \rangle). \tag{39}$$

We assume that this normalizer corresponds to $Z_1$, which is the same assumption as that for the ordinary EP. To calculate $Z_1$, we use the following theorem (its proof is available in Appendix B of the supplementary material):

**Theorem 2** *For the Student-t distribution, we have*

$$\int \exp_t(\langle \Phi(w), \theta \rangle - g) \mathrm{d}w = \left( \exp_t(g_t(\theta)/\Psi - g/\Psi) \right)^{\frac{3-t}{2}}, \tag{40}$$

*where $g$ is a constant, $g(\theta)$ is the log-partition function and $\Psi$ is defined in (7).*

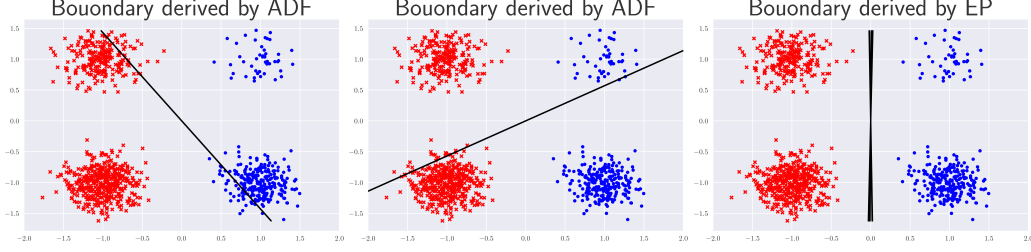

Figure 1: Boundaries obtained by ADF (left two, with different sample orders) and EP (right).

This theorem yields

$$\log_t Z_1^{\frac{2}{3-t}} = g_t(\theta)/\Psi - g_t^{\backslash j}(\theta)/\Psi + \log_t \widetilde{C}_j/\Psi, \tag{41}$$

and therefore the marginal likelihood can be calculated as follows (see Appendix C for details):

$$Z_{\text{EP}} = \int p^{(0)}(w) \otimes_t \prod_i \otimes_t \widetilde{l}_i(w) \mathrm{d}w$$

$$= \left( \exp_t \Big( \sum_i \log_t \widetilde{C}_i/\Psi + g_t(\theta)/\Psi - g_t^{\text{prior}}(\theta)/\Psi \Big) \right)^{\frac{3-t}{2}}. \tag{42}$$

By substituting $\widetilde{C}_i$ in Eq.(42), we obtain the marginal likelihood. Note that, if $t = 1$, the above expression of $Z_{\text{EP}}$ is reduced to the ordinary marginal likelihood expression [9]. Therefore, this marginal likelihood can be regarded as a generalization of the ordinary exponential family marginal likelihood to the t-exponential family.

In Appendices D and E of the supplementary material, we derive specific EP algorithms for the *Bayes point machine* (BPM) and *Student-t process classification*.

## 5   Numerical Experiments

In this section, we numerically illustrate the behavior of our proposed EP applied to BPM and Student-t process classification. Suppose that data $(x_1, y_1), \ldots, (x_n, y_n)$ are given, where $y_i \in \{+1, -1\}$ expresses a class label for covariate $x_i$. We consider a model whose likelihood term can be expressed as

$$l_i(w) = p(y_i | x_i, w) = \epsilon + (1 - 2\epsilon)\Theta(y_i \langle w, x_i \rangle), \tag{43}$$

where $\Theta(x)$ is the step function taking 1 if $x > 0$ and 0 otherwise.

### 5.1   BPM

We compare EP and ADF to confirm that EP does not depend on data permutation. We generate a toy dataset in the following way: 1000 data points $x$ are generated from Gaussian mixture model $0.05N(x; [1, 1]^\top, 0.05I) + 0.25N(x; [-1, 1]^\top, 0.05I) + 0.45N(x; [-1, -1]^\top, 0.05I) + 0.25N(x; [1, -1]^\top, 0.05I)$, where $N(x; \mu, \Sigma)$ denotes the Gaussian density with respect to $x$ with mean $\mu$ and covariance matrix $\Sigma$, and $I$ is the identity matrix. For $x$, we assign label $y = +1$ when $x$ comes from $N(x; [1, 1]^\top, 0.05I)$ or $N(x; [1, -1]^\top, 0.05I)$ and label $y = -1$ when $x$ comes from $N(x; [-1, 1]^\top, 0.05I)$ or $N(x; [-1, -1]^\top, 0.05I)$. We evaluate the dependence of the performance of BPM (see Appendix D of the supplementary material for details) on data permutation.

Fig.1 shows labeled samples by blue and red points, decision boundaries by black lines which are derived from ADF and EP for the Student-t distribution with $v = 10$ by changing data permutations. The top two graphs show obvious dependence on data permutation by ADF (to clarify the dependence on data permutation, we showed the most different boundary in the figure), while the bottom graph exhibits almost no dependence on data permutations by EP.

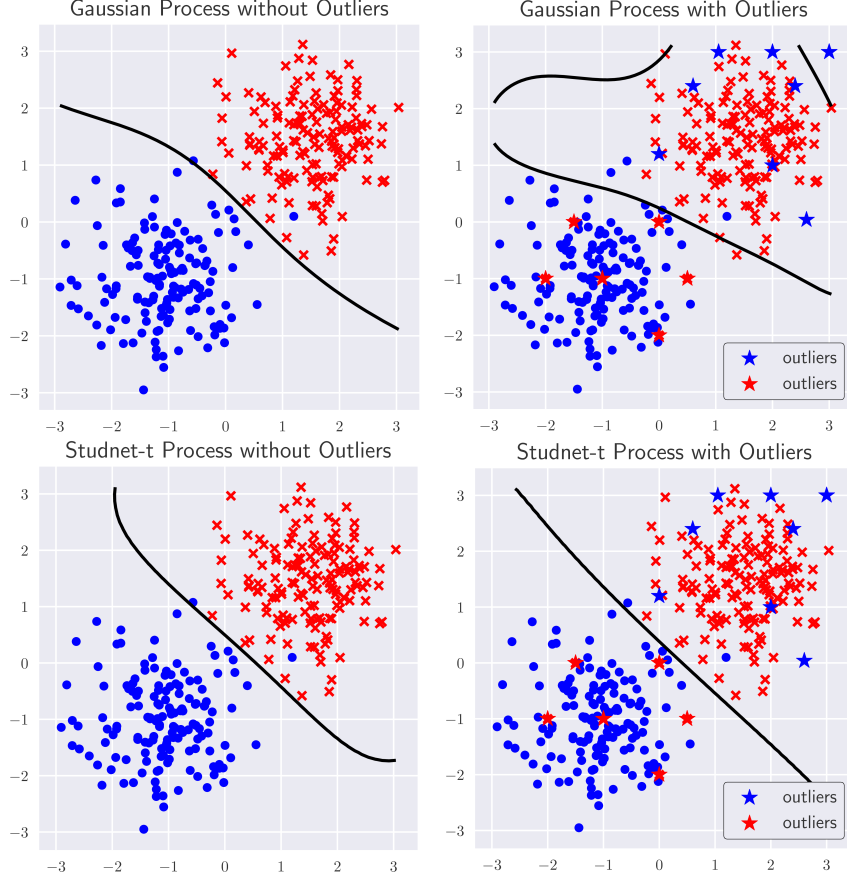

Figure 2: Classification boundaries.

## 5.2 Student-t Process Classification

We compare the robustness of Student-t process classification (STC) and Gaussian process classification (GPC) visually.

We apply our EP method to Student-t process binary classification, where the latent function follows the Student-t process (see Appendix E of the supplementary material for details). We compare this model with Gaussian process binary classification with the likelihood expressed Eq.(43). This kind of model is called robust Gaussian process classification [5]. Since the posterior distribution cannot be obtained analytically even for the Gaussian process, we use EP for the ordinary exponential family to approximate the posterior.

We use a two-dimensional toy dataset, where we generate a two-dimensional data point $x_i$ ($i = 1, \ldots, 300$) following the normal distributions: $p(x|y_i = +1) = N(x; [1.5, 1.5]^\top, 0.5I)$ and $p(x|y_i = -1) = N(x; [-1, -1]^\top, 0.5I)$. We add eight outliers to the dataset and evaluate the robustness against outliers (about 3% outliers). In the experiment, we used $v = 10$ for Student-t processes. We furthermore used the following kernel:

$$k(x_i, x_j) = \theta_0 \exp \left\{ - \sum_{d=1}^{D} \theta_1^d (x_i^d - x_j^d)^2 \right\} + \theta_2 + \theta_3 \delta_{i,j}, \tag{44}$$

where $x_i^d$ is the $d$th element of $x_i$, and $\theta_0, \theta_1, \theta_2, \theta_3$ are hyperparameters to be optimized.

Fig.2 shows the labeled samples by blue and red points, the obtained decision boundaries by black lines, and added outliers by blue and red stars. As we can see, the decision boundaries obtained by the Gaussian process classifier is heavily affected by outliers, while those obtained by the Student-t process classifier are more stable. Thus, as expected, Student-t process classification is more robust

| Table 1: Classification Error Rates (%) | | | |
|---|---|---|---|
| Dataset | Outliers | GPC | STC |
| Pima | 0 | 34.0(3.0) | **32.3(2.6)** |
| | 5% | 34.9(3.1) | **32.9(3.1)** |
| | 10% | 36.2(3.3) | **34.4(3.5)** |
| Ionosphere | 0 | 9.6(1.7) | **7.5(2.0)** |
| | 5% | 9.9(2.8) | **9.6(3.2)** |
| | 10% | 13.0(5.2) | **11.9(5.4)** |
| Thyroid | 0 | **4.3(1.3)** | 4.4(1.3) |
| | 5% | **4.8(1.8)** | 5.5(2.3) |
| | 10% | **5.4(1.4)** | 7.2(3.4) |
| Sonar | 0 | 15.4(3.6) | **15.0(3.2)** |
| | 5% | 18.3(4.4) | **17.5(3.3)** |
| | 10% | **19.4(3.8)** | **19.4(3.1)** |

| Table 2: Approximate log evidence | | | |
|---|---|---|---|
| Dataset | Outliers | GPC | STC |
| Pima | 0 | -74.1(2.4) | -37.1(6.1) |
| | 5% | -77.8(2.9) | -37.2(6.5) |
| | 10% | -78.6(1.8) | -36.8(6.5) |
| Ionosphere | 0 | -59.5(5.2) | -36.9(7.4) |
| | 5% | -75.0(3.6) | -35.8(7.0) |
| | 10% | -90.3(5.2) | -37.4(7.2) |
| Thyroid | 0 | -32.5(1.6) | -41.2(4.3) |
| | 5% | -39.1(2.3) | -45.8(5.5) |
| | 10% | -46.9(1.8) | -45.8(4.5) |
| Sonar | 0 | -55.8(1.2) | -41.6(1.2) |
| | 5% | -59.4(2.5) | -41.3(1.6) |
| | 10% | -65.8(1.1) | -67.8(2.1) |

against outliers compared to Gaussian process classification, thanks to the heavy-tailed structure of the Student-t distribution.

### 5.3 Experiments on the Benchmark dataset

We compared the performance of Gaussian process and Student-t process classification on the UCI datasets[1]. We used the kernel given in Eq.(44). The detailed explanation about experimental settings are given in Appendix F. Results are shown in Tables 1 and 2, where outliers mean how many percentages we randomly flip training dataset labels to make additional outliers. As we can see Student-t process classification outperforms Gaussian process classification in many cases.

## 6 Conclusions

In this work, we enabled the *t-exponential family* to inherit the important property of the exponential family whose calculation can be efficiently performed thorough natural parameters by using the *q-algebra*. With this natural parameter based calculation, we developed EP for the t-exponential family by introducing the *t-factorization* approach. The key concept of our proposed approach is that the t-exponential family has *pseudo additivity*. When $t = 1$, our proposed EP for the t-exponential family is reduced to the original EP for the ordinary exponential family and t-factorization yields the ordinary data-dependent factorization. Therefore, our proposed EP method can be viewed as a generalization of the original EP. Through illustrative experiments, we confirmed that our proposed EP applied to the Bayes point machine can overcome the drawback of ADF, i.e., the proposed EP method is independent of data permutations. We also experimentally illustrated that proposed EP applied to Student-t process classification exhibited high robustness to outliers compared to Gaussian process classification. Experiments on benchmark data also demonstrated superiority of Student-t process.

In our future work, we will further extend the proposed EP method to more general message passing methods or double-loop EP. We would like also to make our method more scalable to large datasets and develop another approximation method such as variational inference.

### Acknowledgement

FF acknowledges support by JST CREST JPMJCR1403 and MS acknowledges support by KAKENHI 17H00757.

## Footnotes

[1] https://archive.ics.uci.edu/ml/index.php

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
