[Supplementary Material]

# Appendix: Expectation Propagation for t-Exponential Family Using q-Algebra

**Futoshi Futami**
The University of Tokyo, RIKEN
futami@ms.k.u-tokyo.ac.jp

**Issei Sato**
The University of Tokyo, RIKEN
sato@k.u-tokyo.ac.jp

**Masashi Sugiyama**
RIKEN, The University of Tokyo
sugi@k.u-tokyo.ac.jp

## A  Proof of Theorem 1

$$
\begin{aligned}
\nabla_{\theta^{\backslash j}} Z_1 &= \nabla_{\theta^{\backslash j}} \int \widetilde{p}^{\backslash j}(w) l_j(w)^t dw \\
&= \int (\phi(w) - \nabla_{\theta^{\backslash j}} g_t(\theta^{\backslash j})) \widetilde{p^{\mathrm{es}}}^{\backslash j}(w) l_j(w)^t dw \\
&= \int \phi(w) \widetilde{p^{\mathrm{es}}}^{\backslash j}(w) l_j(w)^t dw - \nabla_{\theta^{\backslash j}} g_t(\theta^{\backslash j}) \int \widetilde{p^{\mathrm{es}}}^{\backslash j}(w) l_j(w)^t dw
\end{aligned}
$$

Using the definition $Z_2 = \int \widetilde{p^{\mathrm{es}}}^{\backslash j}(w)(l_j(w))^t dw$, and $\eta = \nabla_\theta g_t(\theta)$,

$$
\nabla_{\theta^{\backslash j}} Z_1 = \eta Z_2 - \eta^{\backslash j} Z_2
$$

Therefore,

$$
\eta = \eta^{\backslash j} + \frac{1}{Z_2} \nabla_{\theta^{\backslash j}} Z_1.
$$

## B  Proof of Theorem 2

Here, we consider a one-dimensional case, but we can consider this in the same way as for a multivariate case. Considering the unnormalized t-exponential family, $\exp_t(\langle \Phi(w), \theta \rangle - g)$, and $g$ is a constant, not a true log partition function. We integrate this expression as follows,

$$
\begin{aligned}
\int_{-\infty}^{\infty} \exp_t(\langle \Phi(w), \theta \rangle - g) dw &= \int_{-\infty}^{\infty} (1 + \Psi(-2\mu^\top K w + w^\top K w) - (1-t)g)^{\frac{1}{1-t}} dw \\
&= \int_{-\infty}^{\infty} (1 - \Psi\mu^\top K\mu - (1-t)g + \Psi(w-\mu)^\top K(w-\mu))^{\frac{1}{1-t}} dw \\
&= (1 - \Psi\mu^\top K\mu - (1-t)g)^{\frac{1}{1-t}} \int_{-\infty}^{\infty} \Big(1 + \frac{\Psi(x-\mu)^\top K(x-\mu)}{1 - \Psi\mu^\top K\mu - (1-t)g}\Big)^{\frac{1}{1-t}} dw
\end{aligned}
$$

Here, for simplicity, we put $(1 - \Psi\mu^\top K\mu - (1-t)g) = A$, and use the formula, $\int_0^\infty \frac{x^m}{(1+x^2)^n} dx = \frac{1}{2} B\big(\frac{2n-m-1}{2}, \frac{m+1}{2}\big)$, where $B$ denote the beta function. We can get the expression,

$$
\int_{-\infty}^{\infty} \exp_t(\langle \Phi(w), \theta \rangle - g) dw = \frac{1}{2} B\Big(\frac{3-t}{2(t-1)}, \frac{1}{2}\Big) \Big(\frac{\Psi}{A} K\Big)^{-\frac{1}{2}} A^{\frac{1}{1-t}}
$$

We can proceed with the calculation by using the definition of $\Psi$, $B(x,y) = \frac{\Gamma(x)\Gamma(y)}{\Gamma(x+y)}$, and $\Gamma(\frac{1}{2}) = \sqrt{\pi}$ as follows,

$$\int_{-\infty}^{\infty} \exp_t(\langle \Phi(w), \theta \rangle - g)dw = \Psi^{-\left(\frac{1}{2}+\frac{1}{1-t}\right)} A^{\frac{1}{2}+\frac{1}{1-t}}$$

Here, by using the definition of $A$ and the true log partition function $g_t(\theta) = \frac{1}{1-t}\left(1 - \Psi(\mu^\top K \mu + 1)\right)$,

$$
\begin{aligned}
A^{\frac{1}{2}+\frac{1}{1-t}} &= (1 - \Psi\mu^\top K\mu - (1-t)g)^{\frac{1}{2}+\frac{1}{1-t}} \\
&= (\Psi + (1-t)(g_t(\theta) - g))^{\frac{1}{2}+\frac{1}{1-t}} \\
&= \Psi^{\frac{1}{2}+\frac{1}{1-t}}(1 + (1-t)(g_t(\theta) - g)/\Psi)^{\frac{1}{2}+\frac{1}{1-t}}
\end{aligned}
$$

Therefore, by substituting this expression into the above integral result, we get the following.

$$\int_{-\infty}^{\infty} \exp_t(\langle \Phi(w), \theta \rangle - g)dw = \left(\exp_t(g_t(\theta)/\Psi - g/\Psi)\right)^{\frac{3-t}{2}}$$

## C  Deriving the Marginal likelihood

$$
\begin{aligned}
Z_{\text{EP}} &= \int p^{(0)}(w) \otimes_t \prod_i \otimes_t \widetilde{l}_i(w)dw \\
&= \int \exp_t\left(\sum_i \log_t \widetilde{C}_i + \langle \Phi(w), \theta \rangle - g_t^{\text{prior}}(\theta)\right)dw \\
&= \left(\exp_t\left(\sum_i \log_t \widetilde{C}_i/\Psi + g_t(\theta)/\Psi - g_t^{\text{prior}}(\theta)/\Psi\right)\right)^{\frac{3-t}{2}}.
\end{aligned}
$$

## D  Bayes Point Machine

In this section, we show the details of the update rule of ADF and EP for the Bayes point machine.

### D.1  ADF update rule for BPM

The detailed update rules of ADF for BPM in t-exponential family are derived [1].

$$\mu^i = E_q[w] = \mu^{i-1} + \alpha y_i \Sigma^{i-1} x_i \tag{1}$$

$$\Sigma^i = E_q[ww^\top] - E_q[w]E_q[w^\top] = r\Sigma^{i-1} - (\Sigma^{i-1}x_i)\left(\frac{\alpha y_i \langle x_i, \mu^i \rangle}{x_i^\top \Sigma^{i-1} x_i}\right)(\Sigma^{i-1}x_i)^\top, \tag{2}$$

where $\widetilde{q}_i(w) \propto \widetilde{p}_i(w)^t$, $q_i(w) \propto \widetilde{p}_{i-1}(w)^t(l_i(w))^t$, and

$$z = \frac{y_i \langle x_i, \mu^{i-1} \rangle}{\sqrt{x_i^\top \Sigma^{i-1} x_i}} \tag{3}$$

$$Z_1 = \int \widetilde{p}_{i-1}(w)(l_i(w))^t dw = \epsilon^t + ((1-\epsilon)^t - \epsilon^t)\int_{-\infty}^{z} \text{St}(x; 0, 1, v)dx \tag{4}$$

$$Z_2 = \int \widetilde{q}_{i-1}(w)(l_i(w))^t dw = \epsilon^t + ((1-\epsilon)^t - \epsilon^t)\int_{\infty}^{z} \text{St}(x; 0, v/(v+2), v+2)dx \tag{5}$$

$$r = \frac{Z_1}{Z_2} \tag{6}$$

$$\alpha = \frac{((1-\epsilon)^t - \epsilon^t)\text{St}(z; 0, 1, v)}{Z_2\sqrt{x_i^\top \Sigma^{i-1} x_i}} \tag{7}$$

## D.2 EP update rule for BPM

As for the EP update rule, natural parameters of Student-t distribution $\text{St}(w; v, \mu, \Sigma)$ is $[\theta_1, \theta_2]$,

$$\theta_1 = -2\frac{\Psi K \mu}{1-t} \tag{8}$$

$$\theta_2 = \frac{\Psi K}{1-t} \tag{9}$$

where, $K = (v\Sigma)^{-1}$. From these, we can calculate EP update rules through $\Psi K \mu$ and $\Psi K$.

For the BPM, we consider that the whole approximation is $k$-dimensional $\text{St}(w; m_w, V_w, v)$, and the site approximation as one-dimensional Student-t like function, $\exp_t(\langle \Phi(w), \theta \rangle)$, where $\langle \Phi(w), \theta \rangle = \frac{\Psi_i}{1-t}\left((w^\top x_i)^\top (v\sigma_i)^{-1}(w^\top x_i) - 2m_i(\widetilde{v}\sigma_i)^{-1}(w^\top x_i)\right) \propto \frac{\Psi_i}{1-t}\widetilde{v}^{-1}\sigma_i^{-1}(w^\top x_i - m_i)^2$.

Note that the whole posterior approximation is the $k$-dimensional, but the site approximation is the one-dimensional, therefore the degree of freedom are different from the total approximation and the site approximation to make $t$ consistent. The relation between $v$, $\widetilde{v}$, and $t$ is given as

$$\frac{1}{t-1} = \frac{v+k}{2} = \frac{\widetilde{v}+1}{2}. \tag{10}$$

We denote by $\Psi_i$ and $K_i$ the $\Psi$ and $K$ which is related to site $i$. Here, since $\sigma_i$ is scalar, we ca express $K_i = (\widetilde{v}\sigma_i)^{-1}$. If we express $\Psi = (\alpha/|\Sigma|^{1/2})^{1-t}$, then we ca express $\Psi_i = (\alpha_i/\sigma_i^{1/2})^{1-t}$. We denote by $\Psi_w$ and $K_w$ $\Psi$ and $K$ of whole approximation.

Let us consider the update of site $j$. The first step is calculation of cavity distribution, which can be done by

$$\Psi^{\backslash j} K^{\backslash j} = \Psi_w(vV_w)^{-1} - \Psi_j(\widetilde{v}\sigma_i)^{-1}x_j x_j^\top, \tag{11}$$

$$\Psi^{\backslash j} K^{\backslash j} m^{\backslash j} = \Psi_w(vV_w)^{-1}m_w - \Psi_j(\widetilde{v}\sigma_i)^{-1}m_j x_j. \tag{12}$$

Next step is moment matching. This is calculated in the same way as the ADF update rules. To use the ADF update rule, we have to convert $\Psi^{\backslash j} K^{\backslash j}$ and $\Psi^{\backslash j} K^{\backslash j} m^{\backslash j}$ to $V^{\backslash j}$ and $m^{\backslash j}$, which are covariance matrix and mean of cavity distribution. When calculating $V^{\backslash j}$ from $\Psi^{\backslash j} K^{\backslash j}$, we have to be careful that $\Psi^{\backslash j}$ contains the determinant of $V^{\backslash j}$. From the definition,

$$\Psi^{\backslash j} K^{\backslash j} = \left(\frac{\alpha_j}{|V^{\backslash j}|^{1/2}}\right)^{1-t}(vV^{\backslash j})^{-1}. \tag{13}$$

Since $\alpha_j$ and $v$ is the constant, when we put $\frac{V^{\backslash j^{-1}}}{|V^{\backslash j}|^{(1-t)/2}} = B$, following relation holds,

$$|V^{\backslash j}| = \left(|B|^{\frac{1}{k}}\right)^{\frac{1}{\frac{t-1}{2}-\frac{1}{k}}}. \tag{14}$$

Using this relation, we get $V^{\backslash j}$ and $m^{\backslash j}$.

After moment matching, we get $V_{\text{new}}$ and $m_{\text{new}}$. Next step is the exclusion step of site other than $j$. This step is calculated in the same way as the step of cavity distribution.

$$\Psi_j K_j = \Psi_{\text{new}} K_{\text{new}} - \Psi^{\backslash j} K^{\backslash j}, \tag{15}$$

$$\Psi_j K_j \widetilde{m_j} = \Psi_{\text{new}} K_{\text{new}} m_{\text{new}} - \Psi^{\backslash j} K^{\backslash j} m^{\backslash j}. \tag{16}$$

To update site parameters, we have to convert $\Psi_j K_j$ and $\Psi_j K_j \widetilde{m_j}$ into $\sigma_j$ and $m_j$, which are scalar values. This can be done easily by using the fact that $K_j$ is proportional to $\sigma_j^{-1}x_j x_j^\top$.

These steps are the update rules for the site approximation. We have to iterate these steps until site parameters converge.

# E  Expectation Propagation for Student-t Process Classification

In this section, we show the details of the derivation of EP for the Student-t process classification. The derivation procedure is similar to that of the Gaussian process [5, 4, 2, 3].

## E.1 Deriving Update Rules for Student-t Process Classification

In this subsection, we show the detailed derivation of the update rules for the Student-t process classification. We denote the prior as $p(f|X)$. In the case of Gaussian process, the prior distribution is the multivariate Gaussian distribution whose covariance is specified by the kernel function. In the case of Student-t process, the prior distribution is the multivariate Student-t distribution which is specified by the covariance kernel $k(x,x)$ and the degree of freedom $v$. The posterior distribution is given by $p(f|X,y) = \frac{1}{Z}p(f|X)\prod_i p(y_i|f_i)$, where the marginal likelihood is given as $Z = p(y|X) = \int p(f|X)\prod_i P(y_i|f_i)df$ in the i.i.d. situation. In this paper, we consider a binary classification, and we use

$$p(y_i|f_i) = l_i(f_i) = \epsilon + (1 - 2\epsilon)\Theta(y_i f_i). \tag{17}$$

This is actually the same as BPM, where the input to step function is given as a linear model. In the Student-t process, the input is given as the nonlinear probabilistic process. In this setting, the posterior is intractable. Therefore, we have to approximate it.

Following the EP framework, we approximate the posterior consisting from site approximation. We define the factorizing term that corresponds to data $i$ as follows.

$$\widetilde{l}_i(f_i|\widetilde{C}_i, \widetilde{\mu}_i, \widetilde{\sigma}_i^2) := \widetilde{C}_i \otimes \mathrm{St}(f_i; \widetilde{\mu}_i, \widetilde{\sigma}_i^2, \widetilde{v}) \tag{18}$$

For simplicity, we express the unnormalized Student-t like function by $\mathrm{St}(f_i; \widetilde{\mu}_i, \widetilde{\sigma}_i^2, \widetilde{v})$. This is equivalent to $\exp_t(\langle\Phi(f_i),\theta\rangle)$, where $\langle\Phi(f_i),\theta\rangle = \frac{\Psi_i}{1-t}(f_i^\top K_i f_i - 2\widetilde{\mu}_i^\top K_i f_i) = \frac{\Psi_i}{1-t}(f_i^\top (v\widetilde{\sigma}_i)^{-1}f_i - 2\widetilde{\mu}_i^\top (v\widetilde{\sigma}_i)^{-1}f_i)$. These data corresponding factorizing terms are one-dimensional. Note that the whole posterior approximation is the $k$-dimensional, and site approximation is the one dimensional, the same relation as in the BPM between $v$, $\widetilde{v}$, and $t$ holds as $\frac{1}{t-1} = \frac{v+k}{2} = \frac{\widetilde{v}+1}{2}$.

The q products of this data corresponding term can be expressed as follows:

$$\prod_i \otimes_t \widetilde{l}_i(f_i) = \mathrm{St}(\widetilde{\mu}, \widetilde{\Sigma}, v) \otimes_t \prod_i \otimes_t \widetilde{C}_i \tag{19}$$

Here, we used the property that q products of Student-t distribution become a Student-t distribution. In the above expression, $\widetilde{\mu}$ is the vector of $\widetilde{\mu}_i$ and $\widetilde{\Sigma}$ is the diagonal and following relations are given,

$$\widetilde{K}^{-1} = (v\widetilde{\Sigma}), \tag{20}$$

$$\widetilde{\Psi}\widetilde{K} = \mathrm{diag}(\Psi_1 K_1 \ldots \Psi_n K_n), \tag{21}$$

$$\text{where } \widetilde{\Psi} = \left(\frac{\Gamma((v+k)/2)}{(\pi v)^{k/2}\Gamma(v/2)|\widetilde{\Sigma}|^{1/2}}\cdot\right)^{1-t}. \tag{22}$$

Therefore, the total form of the approximation of the posterior can be expressed as follows.

$$q(f|X,y) = \mathrm{St}(\mu, \Sigma, v) \propto p(f|X) \otimes_t \left(\prod_i \otimes_t \widetilde{l}_i(f_i)\right) \tag{23}$$

From this following relations are obtained,

$$\Psi K = \Psi_0 K_0 + \widetilde{\Psi}\widetilde{K}, \tag{24}$$

$$\Psi K\mu = \widetilde{\Psi}\widetilde{K}\widetilde{\mu}. \tag{25}$$

We consider the case that we update site $i$. For implementation, natural parameter based update rule is preferable. Therefore we define the parameter as follows,

$$\widetilde{\tau}_i = \widetilde{\Psi}_i\widetilde{K}_i, \tag{26}$$

which is the (i,i) element of $\widetilde{\Psi}\widetilde{K}$. We also define,

$$\widetilde{\nu}_i = \widetilde{\Psi}_i\widetilde{K}_i\widetilde{\mu}_i. \tag{27}$$

For the cavity distribution, we define in the same way as,

$$\tau_{-i} = \Psi_{-i}\sigma_{-i}^{-2}\widetilde{v}^{-1}, \tag{28}$$

$$\nu_{-i} = \tau_{-i}\mu_{-i}. \tag{29}$$

The first step is to calculate the cavity distribution, we eliminate the effect of site $i$. To do so, we first integrate out non $i$ terms by using the following formula. Let X and Y are random variable that obey the Student-t distribution,

$$\begin{pmatrix} X \\ Y \end{pmatrix} \sim \mathrm{St}\left( \begin{pmatrix} \mu_x \\ \mu_y \end{pmatrix}, \begin{pmatrix} \Sigma_{xx} & \Sigma_{xy} \\ \Sigma_{yx} & \Sigma_{yy} \end{pmatrix}, v \right). \tag{30}$$

The marginal distribution X is given as,

$$X \sim \mathrm{St}(\mu_x, \Sigma_{xx}, v) \tag{31}$$

By utilizing the above formula, we get

$$q_{-i}(f_i) \quad \propto \quad \int p(f|X) \otimes_t \prod_{j \neq i} \otimes_t l_j(f_j) df_j \tag{32}$$

$$\propto \quad \mathrm{St}(\mu_i, \sigma_i^2, v). \tag{33}$$

where, $\mu_i$ is the $i$th element of $\mu$ and $\sigma_i^2$ is the $(i,i)$ element of $\Sigma$. In the above expression, the degree of freedom is $v$ in both the joint distribution and marginal distribution. This is unfavorable for our Student-t process. To make the EP procedure consistent with $t$, we approximate as $q_{-i}(f_i) \propto \mathrm{St}(\mu_i, \sigma_i'^2, \widetilde{v})$, $\sigma_i'^2 = \sigma_i^2 v / \widetilde{v}$. This is because for a one-dimensional Student-t distribution, its variance is given by $(v\sigma_i^2)^{-1}$, and in this case, $\widetilde{v} > v$, approximation by $\sigma_i'^2 = \sigma_i^2$ would result in the underestimate of the variance.

We calculate the cavity distribution in the following way,

$$\tau_{-i} \quad = \quad \widetilde{v}^{-1} \sigma'^{-2}_i \Psi_i - \widetilde{\tau}_i, \tag{34}$$

$$\nu_{-i} \quad = \quad \widetilde{v}^{-1} \sigma'^{-2}_i \Psi_i \mu_i - \widetilde{\nu}_i. \tag{35}$$

Next step is the inclusion of data $i$ to the approximate posterior. This can be done in the same way of BPM. To derive the update rule, we have to convert $\tau_{-i}$ and $\nu_{-i}$ into $\sigma^2_{-i}$ and $\mu_{-i}$. In this case, the site approximations are one-dimensional, following relation holds,

$$\hat{\mu}_i \quad = \quad \mu_{-i} + \sigma^2_{-i}\alpha, \tag{36}$$

$$\hat{\sigma}_i^2 \quad = \quad \sigma^2_{-i}(r - \alpha\hat{\mu}_i), \tag{37}$$

$$\text{where } \alpha = \frac{((1-\epsilon)^t - \epsilon^t)\mathrm{St}(z:,0,1,\widetilde{v})}{Z_2\sqrt{\sigma^2_{-i}}} \text{ and } z = \frac{y_i\mu_{-i}}{\sqrt{\sigma^2_{-i}}}, \tag{38}$$

where the definition of $Z_2$ and $r$ is same as that of BPM. By using $\sigma^2_{-i}$ and $\mu_{-i}$, we can include the data $i$ information.

After the data inclusion step, we exclude the effect other than data $i$. The calculation of this step can be done in the same way as that of cavity distribution,

$$\widetilde{\tau}_i^{\mathrm{new}} \quad = \quad \widetilde{v}^{-1} \hat{\sigma}_i^{-2} \hat{\Psi}_i - \widetilde{\tau}_{-i}, \tag{39}$$

$$\widetilde{\nu}_i^{\mathrm{new}} \quad = \quad \widetilde{v}^{-1} \hat{\sigma}_i^{-2} \hat{\Psi}_i \hat{\mu}_i - \widetilde{\nu}_{-i}. \tag{40}$$

From this $\widetilde{\tau}_i^{\mathrm{new}}$, we can update $\widetilde{\Psi}\widetilde{K}$. Since $\widetilde{\Psi}\widetilde{K}$ is the diagonal matrix, we just update $(i,i)$ element of $\widetilde{\Psi}\widetilde{K}$.

As a final step, we have to update $\Sigma$. To circumvent the calculation of inverse matrix, we put

$$\Delta\tau = -\widetilde{\tau}_i^{\mathrm{new}} - \widetilde{\tau}_{-i} + \widetilde{v}^{-1}\hat{\sigma}_i^{-2}\hat{\Psi}_i \tag{41}$$

From this, update of $\Psi K$ is given as,

$$\Psi^{\mathrm{new}} K^{\mathrm{new}} = \Psi^{\mathrm{old}} K^{\mathrm{old}} + \Delta\tau e_i e_i^\top \tag{42}$$

where $K^{\mathrm{new}} = (v\Sigma^{\mathrm{new}})^{-1}$ and $K^{\mathrm{old}} = (v\Sigma^{\mathrm{old}})^{-1}$. Here, $\Sigma^{\mathrm{new}}$ is the after the update of $\Sigma$ and $\Sigma^{\mathrm{old}}$ is the before the update of $\Sigma$ and $e_i$ is the unit vector of $i$ th direction. By using the matrix formula, that is, for matrix $A$ and $B$, $(A^{-1} + B^{-1})^{-1} = A - A(A + B)^{-1}A$, we can get the following expression,

$$\Psi^{-1\,\mathrm{new}} v\Sigma^{\mathrm{new}} = \Psi^{-1\,\mathrm{old}} v\Sigma^{\mathrm{old}} - \frac{\Delta\tau}{1 + \Delta\tau \Psi^{-1\,\mathrm{old}} v\Sigma^{\mathrm{old}}} s_i s_i^\top, \tag{43}$$

where $s_i$ is the $i$'s column of $\Psi^{-1\,\mathrm{old}} v\Sigma^{\mathrm{old}}$. From $\Psi^{-1\,\mathrm{new}} v\Sigma^{\mathrm{new}}$, we can get $\Sigma^{\mathrm{new}}$.

These are the update rule of site $i$. We iterate these steps until parameters converge.

### E.2 Hyperparameter Learning

In this subsection, we refer how to derive hyperparameters, such as the wave-length of covariance functions.

In the usual exponential family and Gaussian process, the hyperparameters can be derived by gradient descent for the marginal log likelihood after the EP updates end. Following the discussion [2], we can derive almost the same expression for the gradient of $\log_t Z_{\mathrm{EP}}^{\frac{2}{3-t}}$. When we consider the gradient of hyperparameter $\psi_i$,

$$\frac{\partial \log_t Z_{\mathrm{EP}}^{\frac{2}{3-t}}}{\partial \psi_j} = \eta^\top \frac{\partial \theta_{prior}}{\partial \psi_j} - \eta_{prior}^\top \frac{\partial \theta_{prior}}{\partial \psi_j} + \sum_i \frac{\partial \log_t \widetilde{C}_i}{\partial \psi_j} \tag{44}$$

where, $\theta_{\mathrm{prior}}$ is the natural parameters of prior distribution and $\eta_{\mathrm{prior}}$ is the expected sufficient statistics of the prior distribution.

### E.3 Prediction Rule

In this subsection, we refer to the method of deriving the prediction for the Student-t process classification. After the EP updates end, we will obtain the expression of the approximate posterior distribution as $q(f|X, y) = \mathrm{St}(\mu, \Sigma, v)$.

When a new point $x^*$ is given, we would like to predict its label $y^*$. First we calculate the latent variable $f^*$ of $x^*$. To get the expression of $f^*$, we use the following lemma [6]

**Lemma 1** *If $X \sim \mathrm{St}(\mu, \Sigma, v)$, and $x_1 \in R^{n_1}$, $x_2 \in R^{n_2}$ express the first $n_1$ and remaining $n_2$ entries of X respectively. Then*

$$x_2|x_1 \sim \mathrm{St}\left(\widetilde{\mu}_2, \frac{v + \beta_1}{v + n_1} \times \widetilde{\Sigma}_{22}, v + n_1\right), \tag{45}$$

*where $\widetilde{\mu}_2 = \Sigma_{21}\Sigma_{11}^{-1}(x_1 - \mu_1) + \mu_1$, $\widetilde{\Sigma}_{22} = \Sigma_{22} - \Sigma_{21}\Sigma_{11}^{-1}\Sigma_{12}$, $\beta_1 = (x_1 - \mu_1)^\top K_{11}^{-1}(x_1 - \mu_1)$.*

We consider the following expression,

$$p(\widetilde{f}|X, x^*) = \int p(\widetilde{f}|f, x^*)p(f|X)df. \tag{46}$$

The mean of $p(\widetilde{f}|X, x^*)$ is given by

$$\mathrm{E}[\widetilde{f}] = \int \mathrm{E}[p(\widetilde{f}|f, x^*)]p(f|X)df \tag{47}$$

$$= \int k^\top \Sigma^{-1} f p(f|X)df \tag{48}$$

$$= k^\top \Sigma^{-1}\mu \tag{49}$$

where, $k = [k(x^*, x_1), \ldots k(x^*, x_n)]^\top$. Therefore strict classification of $x^*$ is given by

$$\mathrm{sign}\big(\mathrm{E}[\widetilde{f}]\big) = \mathrm{sign}\big(k^\top \Sigma^{-1}\mu\big) \tag{50}$$

Using this expression, we get the decision boundary.

## F   Experimental setting

We use four datasets from the UCI repository which are widely used for binary classification. We used the cross validation to select the degree of freedom. The range of cross validation for degree of freedom is from 5 to 15.

As discussed in E.2, we derived hyperparameters in the kernel by gradient descent method, especially we used Adam.