[Reviews · NeurIPS 2017]

Reviewer 1



The paper discusses the t-exponential family, and derives an EP scheme for distributions in the t-exponential family. A q-algebra is presented, which allows for computation of the cavity distribution and thus EP. the method is demonstrated on a Bayes point machine and T-process classification. The paper was a great read, really enjoyable, clearly written, fun and engaging. But it's not suitable for NIPS. The emprical validation is not very strong: the first experiment compares EP with ADF, and the result is exactly as expected, and does not really relate to the t-exponential family at all. The second experiment demonstrates that T-process classification is more robust to outliers than GP classification. But the same effect could be acheived using a slight modification to the likelihood in the GP case (see e.g. [1]). There is some value in the work, and the paper is a very enjoyable read, but without a practical upside to the work it remains a novelty. [1] http://mlg.eng.cam.ac.uk/pub/pdf/KimGha08.pdf

Reviewer 2



The authors proposed an expectation propagation algorithm that can work with distributions in the t-exponential family, a generalization of the exponential family that contains the t-distribution as a particular case. The proposed approach is based on using q-algebra operations to work with the pseudo additivity of t-exponential distributions. The gains of the proposed EP algorithm with respect to assumed density filtering are illustrated in experiments with a Bayes point machine. The authors also illustrate the proposed EP algorithms in experiments with a t-process for classification in the presence of outliers. Quality The proposed method seems technically sound, although I did not check the derivations in detail. The experiments illustrate that the proposed method works and it is useful. Clarity The paper is clearly written and easy to follow and understand. I only miss an explanation of how the authors compute the integrals in equation (33) in the experiments. I assume that they have to integrate likelihood factors with respect to Student t distributions. How is this done in practice? Originality The paper is original. Up to my knowledge, this is the first generalization of EP to work with t-exponential families. Significance The proposed method seems to be highly significant, extending the applicability of EP with analytic operations to a wider family of distributions, besides the exponential one.

Reviewer 3



The authors propose to use q-algebra to extend the EP method to t-exponential family. More precisely, based on the pseudo additivity of deformed exponential functions which is commonly observed in t-exponential family, the authors exploit a known concept in statistical physics called q-algebra and its properties such as q-product, q-division, and q-logarithm, in order to apply these properties to derive EP for t-exponential family. The paper is well written and easy to follow (although I haven't thoroughly checked their proofs for Theorems). I find the experiments section a bit unexciting because they showed known phenomena, which are (1) ADF depends on data permutation and EP doesn't, and (2) GP classification is more vulnerable to outliers than student-t process. But I would guess the authors chose to show these since there is no existing methods that enables EP to work for t-exponential family. --- after reading the author's rebuttal --- Thanks the authors for their explanations. I keep my rating the same.